# Recovering *BRAF* Signal from Individual Tumor Patches in Papillary Thyroid Carcinoma H&E Whole-Slide Images

**YongHun Lee**[1] (iD)                                                   CODE@UNIST.AC.KR
**Jiwon Song**[2]                                                WAYNE0525@UNIST.AC.KR
**JuHyeong Ki**[3]                                                 JUHYEONGKI@UNIST.AC.KR
**YongKyung Oh**[4]                                        YONGKYUNGOH@MEDNET.UCLA.EDU
**HaYoung Lee**[5]                                          DLGKDUADDL563@NCC.RE.KR
**EunKyung Lee**[5]                                                   EKLEE@NCC.RE.KR
**Jimin Lee**[1,3]                                              JIMINLEE@UNIST.AC.KR

[1] *Artificial Intelligence Graduate School, Ulsan National Institute of Science and Technology, Ulsan, Republic of Korea*

[2] *Department of Biomedical Engineering, Ulsan National Institute of Science and Technology, Ulsan, Republic of Korea*

[3] *Department of Nuclear Engineering, Ulsan National Institute of Science and Technology, Ulsan, Republic of Korea*

[4] *Medical & Imaging Informatics (MII) group, University of California, Los Angeles, USA*

[5] *Center for Thyroid Cancer, National Cancer Center, Goyang, Republic of Korea*

## Abstract

*BRAF* V600E mutation has important diagnostic and prognostic implications in papillary thyroid carcinoma (PTC), and predicting its status from H&E whole-slide images could replace ancillary testing for *BRAF* assessment in selected settings. In this study, we found that tumor patches from $BRAF^{\mathrm{wt}}$ and $BRAF^{\mathrm{mt}}$ cases were distinguishable at the patch level, and that this difference remained evident after aggregation to the slide level. To examine this, we performed a controlled comparison of patch-level feature representations within a tumor-focused pipeline using a curated PTC H&E WSI cohort of 665 patients with matched *BRAF* labels. Pathology foundation models (UNI2 and Virchow2) showed higher performance than an ImageNet-pretrained ResNet50 under identical downstream settings. Overall, the results support the presence of local morphologic patterns linked to *BRAF* status in PTC H&E morphology.

**Keywords:** Whole-Slide Imaging, *BRAF* Mutation, Representation Learning, Computational Pathology

## 1. Introduction

*BRAF* V600E mutation is a key molecular alteration in papillary thyroid carcinoma (PTC), and predicting its status from hematoxylin and eosin (H&E) whole-slide images (WSIs) could replace ancillary testing such as IHC or PCR for initial *BRAF* assessment in selected settings. Prior work in PTC has already established the feasibility of predicting genetic alterations, including *BRAF* status, from H&E using an ImageNet-pretrained Vision Transformer (Marion et al., 2025).

Rather than re-establishing this feasibility, we ask whether case-level *BRAF* signal is recoverable from individual tumor patches and how strongly this recovery depends on the

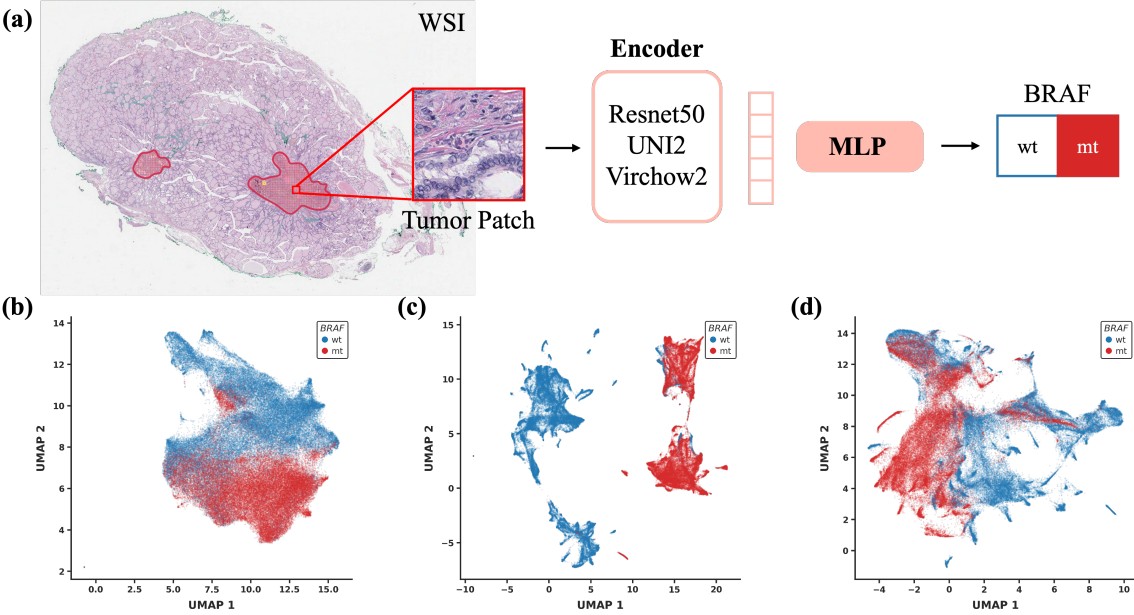

Figure 1: Tumor-focused pipeline and UMAP projections of patch embeddings. (a) Tumor-focused pipeline. (b) ResNet50. (c) UNI2. (d) Virchow2.

underlying feature representation. Using a curated single-institution cohort of 665 PTC patients with matched $BRAF$ labels, we compare ImageNet-pretrained ResNet50 features with pathology foundation model features from UNI2 (Chen et al., 2024) and Virchow2 (Zimmermann et al., 2024) within the same tumor-focused pipeline. We evaluate both patch-level prediction and the corresponding aggregated slide-level predictions.

## 2. Methods

We curated a single-institution PTC H&E WSI cohort from postoperative pathology reports at the National Cancer Center, comprising 665 patients with matched $BRAF$ labels, including 214 $BRAF^{\text{wt}}$ and 451 $BRAF^{\text{mt}}$ cases. Cases were labeled as $BRAF^{\text{wt}}$ (wildtype) or $BRAF^{\text{mt}}$ (mutant), with the latter assigned when either IHC or PCR was positive. Tumor regions were annotated by outlining the full mass or nodule, including infiltrative margins, and all annotations were validated by a pathologist. Data were split at the patient level into training, validation, and test sets of 384, 139, and 142 cases, with $BRAF^{\text{wt}}/BRAF^{\text{mt}}$ counts of 74/310, 69/70, and 71/71, respectively. $256 \times 256$ patches were extracted from the tumor regions for downstream representation learning. For patch-level training and evaluation, all tumor patches extracted from a slide inherited the corresponding slide-level $BRAF$ label.

Within the same tumor-focused pipeline (Figure 1), we compared frozen ResNet50, UNI2, and Virchow2 encoders under identical downstream settings. Performance was evaluated at both the patch and slide levels using AUROC and accuracy. Slide-level predictions were obtained by majority voting over patch-level predictions.

Table 1: Patch- and slide-level performance under matched downstream settings.

| Model | Patch-AUROC | Patch-Acc | Slide-AUROC | Slide-Acc |
|---|---|---|---|---|
| ResNet50 | 0.9695 | 0.9258 | 0.9825 | 0.9296 |
| UNI2 | 0.9840 | **0.9537** | **0.9919** | **0.9507** |
| Virchow2 | **0.9842** | 0.9528 | 0.9903 | 0.9437 |

## 3. Experiments

All encoders were evaluated within the same tumor-focused pipeline under identical downstream settings, with a separate MLP classifier trained for each encoder. In the test set, the $BRAF^{\text{wt}}$ and $BRAF^{\text{mt}}$ groups each comprised 71 patients, corresponding to 67,714 and 48,217 tumor patches, respectively. Performance was assessed at both the patch and slide levels using AUROC and accuracy, with slide-level predictions obtained by majority voting over patch-level predictions.

Table 1 summarizes patch-level and slide-level performance for all three encoders. Across all encoders, case-level $BRAF$ signal was recoverable from individual tumor patches, and this pattern remained evident after aggregation to the slide level. The pathology foundation models yielded consistently higher performance than the ImageNet-pretrained baseline at both levels, with UNI2 showing the strongest overall slide-level results. These results support the view that PTC H&E morphology contains local signal relevant to case-level $BRAF$ status, while also suggesting that some feature representations recover this signal more effectively than others.

Figure 1 shows broader overlap between $BRAF^{\text{wt}}$ and $BRAF^{\text{mt}}$ patches for ResNet50, clearer separation for UNI2, and an intermediate pattern for Virchow2. This qualitative trend is consistent with the quantitative results and further supports the presence of local morphologic patterns associated with case-level $BRAF$ status.

## 4. Conclusion

These findings support the potential clinical value of routine histology as an accessible source of information for early case-level $BRAF$ assessment in PTC and, in selected settings, as a potential substitute for ancillary testing. The comparative analysis further suggested that the strength of this signal depends on the underlying feature representation. Future work toward clinical translation should explore whether IHC-derived supervision can improve H&E representations while preserving H&E-only inference for practical $BRAF$ assessment.

## Acknowledgments

This study was supported by research funds of National Cancer Center (No. 2510400-2, 2410863-3).

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
