# OpenReview forum: "Recovering BRAF Signal from Individual Tumor Patches in Papillary Thyroid Carcinoma H&E Whole-Slide Images"
_MIDL.io/2026/Short_Papers — MIDL 2026 - Short Papers Poster_

### Official Review · Reviewer_sAjs · 2026-05-03

**Rating:** 4
**Confidence:** 4

**Review:**

This work presents a clear and focused study addressing an interesting question: whether molecular signals (BRAF mutation) can be recovered at the patch level and how representation learning affects this. The experimental design is well controlled, with identical downstream pipelines across encoders, making comparisons fair and interpretable. Results are strong and consistent, though the contribution is somewhat incremental relative to prior work on mutation prediction from histopathology.

**Summary:**

This paper investigates whether case-level BRAF mutation status in papillary thyroid carcinoma can be recovered from individual tumor patches extracted from H&E whole-slide images, rather than relying solely on slide-level aggregation. The authors design a tumor-focused pipeline and perform a controlled comparison of feature representations using ImageNet-pretrained ResNet50 and pathology foundation models (UNI2 and Virchow2) on a curated cohort of 665 patients. Their experiments demonstrate that patch-level signals are highly predictive of BRAF status and remain consistent after aggregation to slide-level predictions. Furthermore, they show that pathology foundation models outperform standard ImageNet features, suggesting that representation choice significantly impacts the recoverability of molecular signals from histopathology. These findings highlight the presence of localized morphological correlates of BRAF mutation and support the potential of foundation models in computational pathology.

**Strengths:**

1. The paper addresses a well-defined and relevant research question, moving beyond feasibility toward understanding where predictive signal resides (patch vs. slide level).
2. The experimental setup is carefully controlled, with identical downstream pipelines ensuring fair comparison across feature representations.
3. The comparison between ImageNet-pretrained models and pathology foundation models is timely and demonstrates clear performance gains.

**Weaknesses:**

1. The contribution is somewhat incremental, as prior work has already demonstrated mutation prediction from histopathology; the novelty mainly lies in the patch-level analysis.
2. Patch-level labeling inherits slide-level labels, which may introduce label noise and overestimate patch-level discriminability.
3. The aggregation strategy (majority voting) is relatively simple and does not explore more advanced multiple instance learning approaches.

**Justification Of Rating:**

This is a solid and well-executed short paper that provides clear empirical insights into the role of patch-level representations in mutation prediction. While the novelty is moderate and some methodological aspects could be further explored, the study is rigorous, clearly written, and relevant to the MIDL community. Given the short paper track’s goal of inclusivity and dissemination, and the absence of major flaws, I recommend acceptance.

---

### Decision · Program_Chairs · 2026-05-08

Accept (Poster)